

# Efficient polynomial analysis of MAS spinning sidebands and application to order parameter determination in anisotropic samples

Günter Hempel[1], Paul Sotta[2], Didier R. Long[3], and Kay Saalwächter[1]

[1]Martin-Luther-Universität Halle-Wittenberg, Institut für Physik – NMR, Betty-Heimann-Str. 7, 06120 Halle, Germany
[2]Ingénierie des Matériaux Polymères, INSA Lyon/CNRS UMR 5223, 17 avenue Jean Capelle, 69621 Villeurbanne cedex, France
[3]Université Lyon, INSA Lyon/CNRS, UCBL, MATEIS, UMR5510, 69100 Villeurbanne, France

**Correspondence:** Günter Hempel (guenter.hempel@physik.uni-halle.de)

**Abstract.** Chemical shift tensors in $^{13}$C solid-state NMR provide valuable localized information on the chemical bonding environment in organic matter, and deviations from isotropic static-limit powder lineshapes sensitively encode dynamic-averaging or orientation effects. Studies in $^{13}$C natural abundance require magic-angle spinning (MAS), where the analysis must thus focus on spinning sidebands. We propose an alternative fitting procedure for spinning sidebands based upon a polynomial expansion that is more efficient than the common numerical solution of the powder average. The approach plays out its advantages in the determination of CST principal values from spinning-sideband intensities and order parameters in non-isotropic samples, which is here illustrated on the example of stretched glassy polycarbonate.

## 1 Introduction

The chemical-shift anisotropy (CSA) is one of the most useful interactions in solid-state NMR, as the principal values of its tensor span a convenient frequency range for many relevant heteronuclei present in organic materials, such as $^{13}$C, $^{15}$N or $^{31}$P. Excluding effects of intermediate motions on the NMR timescale, deviations from the static-limit isotropic powder lineshapes, characterized by the 3 principal values or the 3 commonly derived invariants (isotropic shift, anisotropy and asymmetry), are immediately informative about the geometry of fast-limit motions (Kulik et al., 1994; Titman et al., 1994) as well as orientation effects in non-isotropic samples (Maricq and Waugh, 1979; Hentschel et al., 1978). The latter are the main concern of this contribution.

The most complete information would be the extraction of the full orientation distribution function (ODF), which is best achieved with the dedicated "DECODER" 2D experiment involving a mechanical sample flip (Schmidt-Rohr et al., 1992), or with some compromises in special cases even from 1D spectra (Hempel and Schneider, 1982). Alternatively, the anisotropy can be quantified by orientational moments, which are proportional to expansion coefficients of the orientation distribution in terms of Legendre polynomials. For the evaluation of static powder lineshapes, two different schemes for the estimation of orientational moments were applied, (i) a decomposition of the spectra into elementary spectra belonging to different Legendre polynomials (Hentschel et al., 1978), or (ii) the estimation of the orientational moments from the lineshape moments (Hempel et al., 1999). All these low-resolution approaches suffer from spectral overlap, leaving selective isotope labeling, possibly also





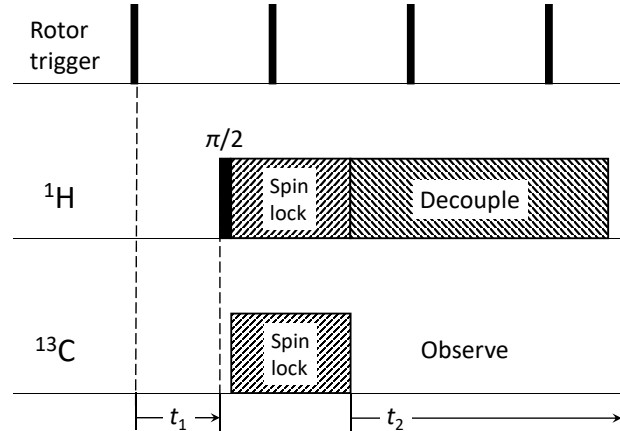

**Figure 1.** Pulse sequence for 2D syncMAS (Harbison et al., 1987). The direct dimension consists of the usual data acquisition time domain after cross polarization, while the indirect dimension consists in an incrementation of the delay between a rotor trigger and the start of the actual pulse sequence, covering a single rotor period $T_r$ typically following $t_1 = i\,T_r/2^n$ with $i = 0\ldots 2^n$ ($2^n$=16 in our case).

with $^2$H (Spiess, 1982) and harnessing its well-defined quadrupolar coupling, as a certainly non-routine and often unfeasible

alternative.

Multiple sites even in natural abundance of $^{13}$C can of course be addressed in single experiments using fast magic-angle spinning (MAS), but at the expense of removing the anisotropy effects from the spectra. One must then rely on recoupling experiments such as CODEX (deAzevedo et al., 2000) or the more dedicated SUPER experiment (Liu et al., 2002), but these are often subject to uncertainties related to scaling factors and line-broadening arising from experimental imperfections. In

this paper, we demonstrate that a somewhat "revisionist" approach of using lower spinning frequencies and the focus on a spinning sideband (SSB) manifold can help to solve a given problem without strong requirements concerning the sample or the spectrometer hardware. A simple comparison of sideband intensites of an isotropic *vs.* an oriented sample may be sufficient. With some more effort in terms of experimental time, one can record a series of spectra with incremented triggered rotor phase, resulting in the 2D "syncMAS" experiment (Harbison et al., 1987). which enables much better accuracy (see Fig. 1). A further

improvement in line separation is provided by the time-consuming 3D ORDER method (Titman et al., 1993), where spinning sidebands of different order a separated in different 2D planes of the 3D stack.

In any such experiment, precise knowledge of the CS tensor is required, which can again be deduced from MAS sidebands. A famous analytical relation between tensor and SSB intensities was given by Herzfeld and Berger (1980). However, their equation is far too complicated to be applied. Instead, computer programs involving numerical solutions of the powder average

integral are nowadays available and readily applicable. One can proceed along this line and obtain orientational moments in an anisotropic sample by numerical calculation of SSB subspectra. At this point, we argue that a more simple analytical connection between SSB intensities and anisotropy parameters would be very helpful, for instance in the form of polynomials. Then, well-etablished fitting procedures (such as Gauss-Newton, Levenberg-Marquardt and others) could be applied, also





for the estimation of the uncertainties. The aims of this paper are as follows: (i) The introduction of an exact polynomial
representation of SSB intensities. This is tested by evaluating SSB in glycine, for which the CSA principle values are known
from single-crystal measurements. (ii) The derivation of a sixth-order polynomial approximation of syncMAS NMR data. This
is demonstrated to be of great use in estimating the orientational moments in stretched glassy polycarbonate (PC). Our data
analysis approach offers more flexibility with regards to the inherent model assumptions, which cannot as easily be tested
or changed within the originally proposed data analysis scheme relying on pre-calculated subspectra (Harbison et al., 1987).
It goes without saying that our results are readily generalized for the case of dipole-dipole interactions (for heteronuclear or
isolated homonuclear spin pairs), and also for first-order quadrupolar interactions ($^2$H, $^7$Li).

## 2 Theoretical part

### 2.1 Definitions

#### 2.1.1 CSA tensor parameters

In solid-state NMR, the anisotropic electronic shielding effect is written as a dimensionless tensor $\underline{\sigma}$,

$$\mathbf{B}_{\mathrm{loc}} = (1 - \underline{\sigma})\mathbf{B}_0 \, , \tag{1}$$

where $\mathbf{B}_0$ the external magnetic field and $\mathbf{B}_{\mathrm{loc}}$ the local field at the position of the nucleus. The shielding effect is always
referenced to a known isotropic shift of a reference compound $\sigma_{\mathrm{ref}}$ (**1**: unit tensor),

$$\underline{\delta} := \underline{\sigma} - \sigma_{\mathrm{ref}} \cdot \mathbf{1} \, . \tag{2}$$

The tensor, henceforth referred to as CST (chemical-shift tensor), has only real eigenvalues and is uniquely defined by 6 inde-
pendent quantities, where one commonly reports 3 eigenvalues and 3 Euler angles, the latter characterizing the orientation of
the principal-axes frame (PAF). For an isotropic static powder sample or a MAS sideband manifold the orientation information
is lost and one can only measure the 3 eigenvalues. We follow the common convention for the principal components: $\delta_{33}$ is the
eigenvalue which deviates most from the isotropic shift $\delta_{\mathrm{iso}} := (\delta_{11} + \delta_{22} + \delta_{33})/3$, and $\delta_{22}$ deviates least. Alternatively, one
can also use only the invariants: isotropic shift $\delta_{\mathrm{iso}}$, anisotropy $\delta := \delta_{33} - \delta_{\mathrm{iso}}$, and asymmetry parameter $\eta := (\delta_{22} - \delta_{11})/\delta$.
With this we have two possiblities: $\delta_{33} < \delta_{\mathrm{iso}} \leq \delta_{22} \leq \delta_{11}$ and $\delta$ is positive, or $\delta_{33} > \delta_{\mathrm{iso}} \geq \delta_{22} \geq \delta_{11}$ and $\delta$ is negative. In either
case we fulfil the convention $0 \leq \eta \leq 1$. For sake of simplicity we assume in the following that the spectrometer frequency is
set to the center of the powder pattern, i.e. $\delta_{\mathrm{iso}} := 0$.

We finally comment on an often neglected aspect: As there are 6 possibilities to assign the three eigenvalues to three principal
axes of the CST, there are 6 solutions for the pair $\{\delta, \eta\}$. Exchange of the values for $\delta_{11}$ and $\delta_{22}$ simply changes the sign of $\eta$ but
cyclic permutation of the indices produces more complex changes. The resulting 6 value pairs all yield the identical static-limit
powder spectrum or SSB pattern. Only one of them fulfills $0 \leq \eta \leq 1$, but some cases exist where it might be helpful to deviate
from this convention. One example is discussed below, where the CST of para-substituted phenylene carbons will be assigned





in different ways to the PAF. For sake of simplicity, it will be advantageous to surrender the numbering order from above for
one of the carbons; the benefit will be a common frame for both carbons which simplifies the data evaluation appreciably.

### 2.1.2   Angle conventions, transformations and orientational moments

The focus of the second part is on the description of orientational effects of molecular-scale structural units characterized by
a given distribution of orientations. Follwoing Harbison et al. (1987) and Schmidt-Rohr and Spiess (1994), we summarize the
relevant definitions. Starting with the common transformation from the PAF to the mlecular frame (which we will later identify
with a main-chain section of the polymer backbone), we need an additional frame that is related to the macroscopic sample
deformation ("director frame"). The order of the required transformations is:

PAF → molecular frame → director frame → MAS rotor frame → lab frame.

For the purpose of symbolic treatment, two changes are made to simplify the resulting expressions as much as possible.
For the background of both arguments, we refer to Hentschel et al. (1978); Harbison et al. (1987); Henrichs (1987). First, the
orientation of a frame with respect to its preceding one is characterized by three Euler angles. Following this scheme, for the
transformation from one of the frames above to the next, three single-angle rotations are required. It is well-known property of
sequential Euler rotations that the first single-angle rotation of a succeeding transformation is simply the continuation of the
third single-angle rotation of the preceding transformation, i.e. both rotations are performed around the same axis. Particularly
for symbolic treatment the result will be simplified substantially if both equal-axis rotations are combined to a single rotation
by the sum angle. This simplifies the problem to only two rotations per transformation between succeeding frames, which are
(1) rotation around the $z$ axis such that the $y$ axis is parallel to the new $y$ axis and both $xz$ planes are parallel, and (2) rotation
around the $y$ axis to reach the new frame. This results in a sequence of 4 double rotations alternating around $z$ and $y$ axes
instead of 4 triple transformations by the complete sets of Euler angles.

Second, the coordinate transformations are usually performed in Cartesian vector space, i.e., the 3x3 matrix of the CST is
multiplied bilinearly from left as well as from right with 3x3 matrices. Also here, a possibility for simplification is used which
is in the spirit of using a spherical representation that relies on linear combination using Wigner matrix elements, but which
is defined in Cartesian space. We use symmetric matrices with 6 independent elements instead of 9 ones in the general case,
which do not require the full set of operations. Instead of the bilinear matrix operations, we rather use transformations in tensor
space (details are to be published under separate cover), in which the traceless part of the CST is represented by a 5-membered
column vector. The transformation matrices have the size 5x5, but have to be applied only once, from left. Matrices for $z$
rotation and $y$ rotation by angle $\psi$ are denoted by $\mathbf{R}_z(\psi)$ and $\mathbf{R}_y(\psi)$, respectively.

The following angles are relevant:

- CST PAF to molecular frame: $z$ rotation by $\psi$ (azimuth) and $y$ rotation by $\alpha$ (polar angle)

- molecular frame to director frame: $z$ rotation by $\varepsilon$ and $y$ rotation by $\beta$

- director frame to rotor frame: $z$ rotation by $\varphi$ and $y$ rotation by $\beta_2$





— rotor frame to lab frame: $z$ rotation by $\gamma$ and $y$ rotation by the magic angle $\vartheta_{\mathrm{MA}} = \arccos(\frac{1}{\sqrt{3}})$

With these definitions we can move to the specific features of the given problem. Following Roe (1970) we have an ODF $W(\varepsilon, \cos\beta, \varphi)$ of Euler angles. This function can be expanded in terms of Wigner matrices; the determination of the expansion coefficients is the goal of our syncMAS experiments (see Fig. 1). For uniaxial deformation the ODF depends on the polar angle $\beta$ only, and not on the azimuth $\varphi$. In this case there is no preferred lateral orientation of the molecular units with respect to the plane spanned by the $z$ axes of molecular frame and the director frame, which means that all $\varepsilon$ have equal probability. It is then sufficient to describe orientation effects by a 1D uniaxial function $U(\cos\beta)$. It can be expanded in terms of Legendre polynomials $P_l(\cos\beta)$,

$$U(\cos\beta) = \sum_{n=0}^{\infty} C_n\, P_n(\cos\beta)\ ; \quad n \in \mathbb{N}\ . \tag{3}$$

The expansion coefficients are

$$C_n = \frac{1}{2n+1}\langle P_n\rangle \quad \text{with} \quad \langle P_n\rangle := \int_0^1 P_n(\cos\beta)\, U(\cos\beta)\, \mathrm{d}\cos\beta\ ; \quad n \in \{1, 2, ..\}\ . \tag{4}$$

According to Henrichs (1987) we denote the $\langle P_n\rangle$ as orientational moments. NMR methods are sensitive only to the symmetric part of the ODF of the CSTs, $U(\cos\beta)$; any non-zero skew-symmetric parts cannot be detected by evaluating CSA spectra; hence all odd orientational moments vanish.

## 2.2 Calculation procedure

The treatments of the 1D MAS and the 2D syncMAS experiments are largely equivalent, and we here summarize the sequence of calculation steps.

1. Estimation of the angle $\Phi$ ("phase") between the instantaneous magnetization direction and the initial direction by time integration of the instantaneous precession frequency $\omega(t)$;

$$\Phi(t) = \int_0^t \omega(t')\, \mathrm{d}t' \tag{5}$$

$\omega$ and therefore $\Phi$ will depend on the orientation of the CST with respect to $\mathbf{B_0}$, which depends periodically on time due to MAS. The angles which describe the tensor orientation are chosen such that the time dependence is contained in one angle termed rotor angle $\gamma$ specifying the instantaneous rotor position.

For 1D MAS:

$$\gamma(t) = \omega_r t + \gamma_0 \tag{6}$$

For 2D syncMAS:

$$\gamma(t_1, t_2) = \omega_r t_2 + \gamma_0(t_1)\quad ; \quad \gamma_0(t_1) = \omega_r t_1 + \gamma_{00} \tag{7}$$



$\gamma_0$ describes the rotor position at the end of signal excitation (= start of the data acquisition) of the current experiment; $\gamma_{00}$ describes the rotor position at the start of acquisition of the very first of the 2D slices. In case of 1D MAS, we used the well-known equations from the literature for $\Phi(t)$, see the next section. For describing the 2D experiment, an equivalent analytical expression for the instantaneous precession frequency is easily derived and integrated.

2. Calculation of orientational averages of phase powers $\langle\Phi^n(t)\rangle_{\mathrm{or}}$. In the particular case of an isotropic sample, this average is the powder average $\langle\Phi^n(t)\rangle_{\mathrm{powder}}$.

3. Assembling the FID and estimation of the SSB intensities by Fourier analysis *via*

$$FID(t) = \left\langle e^{i\Phi(t)} \right\rangle_{\mathrm{or}} = \sum_{n=0}^{\infty} \frac{i^n}{n!} \langle\Phi^n(t)\rangle_{\mathrm{or}} \tag{8}$$

The periodicity of the MAS signal originates from the periodic modulation of the precession frequency. The integration providing the phase generally gives the sum of a likewise periodic component and a linear component. If the angle between rotation axis and $\mathbf{B_0}$ is exactly the magic angle $\arccos\frac{1}{\sqrt{3}}$, and if the spectrometer frequency is set to the isotropic average of the CS, the linear term vanishes and $\Phi$ is a purely periodic function. This further holds for the orientation-averaged phase powers. Therefore, for physical reasons we expect periodic FIDs which can be written as Fourier series:

$$FID(t) = \begin{cases} \sum\limits_{m=-\infty}^{\infty} I_m\, e^{im\omega_r t} & \text{for 1D MAS} \\ \sum\limits_{m=-\infty}^{\infty} \sum\limits_{k=-\infty}^{\infty} I_{mk}\, e^{im\omega_r t_2}\, e^{ik\omega_r t_1} & \text{for 2D syncMAS} \end{cases} \tag{9}$$

After FT, the $I_m$ appear as intensities of the SSB in the 1D MAS spectrum and the $I_{mk}$ as intensities of the 2D SSB in the 2D syncMAS spectrum.

## 2.3 Polynomials for 1D SSB intensities of an isotropic sample

Step 1 (phase):
For this case we can neglect the intermediate transformations involving the molecular frame and the director frame, and use a single transformtion from the CST PAF directly into the rotor frame using only the angles $\psi$ (azimuth) and $\alpha$ (polar angle). In the absence of thermal motion, the time dependence due to the motion of the tensor under MAS leads to (Schmidt-Rohr and Spiess, 1994; Duer, 2002):

$$\Phi(t) = \frac{\omega_0\,\delta}{\omega_r}\,\frac{1}{12}\Big\{4\sqrt{2}\,\eta\sin\alpha\sin2\psi\,f_c(t) + 2\eta\cos\alpha\sin2\psi\,f_{2c}(t)$$
$$-2\sqrt{2}\,(3-\eta\,\cos2\psi)\sin2\alpha\,f_s(t) + \Big[3\sin^2\alpha + \frac{\eta}{2}\,(3+\cos2\alpha)\cos2\psi\Big]f_{2s}(t)\Big\} \tag{10}$$

with the abbreviations

$$f_c(t) := \cos(\omega_r t + \gamma_0) - \cos\gamma_0\,; \qquad f_{2c}(t) := \cos(2\omega_r t + 2\gamma_0) - \cos2\gamma_0\,;$$

$$f_s(t) := \sin(\omega_r t + \gamma_0) - \sin\gamma_0\,; \qquad f_{2s}(t) := \sin(2\omega_r t + 2\gamma_0) - \sin2\gamma_0\,. \tag{11}$$





$\omega_0 = 2\pi f_0$ with $f_0$ being the Larmor frequency and $\omega_\mathrm{r}$ is the spinning rate in units of angular frequency.


Step 2 (orientational averages of phase powers):

Steps 2 and 3 could be performed by inserting these expressions into equation (8) by symbolic software (here Mathematica) for $n \leq 14$. These are the same expressions as listed below. However, a general expression was not found in this way. To obtain such a general expression for $\Phi^n$ with $n \in N$ to obtain terms of arbitrarily high order, we factorize $\langle \Phi^n \rangle$ into a term which

depends only on time and one which depends only on orientation. The separation of time and orientation dependence enables symbolic calculations. This can be achieved by replacing

$$f_c(t) = \cos(\omega_\mathrm{r} t + \gamma_0) - \cos\gamma_0 \quad \rightarrow \quad -2\sin\gamma_2 \, \sin\frac{\omega_\mathrm{r} t}{2} \quad \text{with } \gamma_2 := \frac{\omega_\mathrm{r} t}{2} + \gamma_0 \tag{12}$$

and similarily for $f_s(t)$, $f_{2c}(t)$ and $f_{2s}(t)$.

The phase powers can be written as

$$\Phi^n = \left( \frac{\omega_0 \, \delta}{\omega_r} \right)^n \left( A\sin\frac{\omega_r t}{2} + B\sin\omega_r t \right)^n \tag{13}$$

with

$$
\begin{aligned}
A :&= -\frac{2\sqrt{2}}{3} \left[ \eta \, \sin\gamma_2 \, \sin\alpha \, \sin 2\psi + \frac{1}{2} \cos\gamma_2 \, \sin 2\alpha \, (3 - \eta \, \cos 2\psi) \right] \\
B :&= \frac{1}{3} \left\{ -\eta \, \sin 2\gamma_2 \, \cos\alpha \, \sin 2\psi + \frac{1}{2} \cos 2\gamma_2 \left[ \eta \left( 1 + \cos^2\alpha \right) \cos 2\psi + 3\sin^2\alpha \right] \right\}
\end{aligned} \tag{14}
$$

$\gamma_2$ describes the rotor position in the middle of the integration interval $[0,t]$ in eq. (5). It can be regarded as an azimuthal angle which can be used for a powder average. Therefore, $A$ and $B$ are effectively time-invariant. In combination with the binomial

law we can convert eqn. (13) to

$$\Phi^n = \left( \frac{\omega_0 \, \delta}{\omega_r} \right)^n \sum_{k=0}^{[n/2]} \binom{n}{2k} A^{2k} B^{n-2k} \sin^{2k}\frac{\omega_r t}{2} \sin^{n-2k}\omega_r t . \tag{15}$$

Here we made use of the fact that powder averages with odd powers of $A$ vanish. $[n/2]$ denotes the integer part of $n/2$.

We now have a sum of products in which orientation- and time-dependent terms are separated into separate factors. Thus, powder average can be restricted to $A^{2k} B^{n-2k}$:

$$\left\langle A^{2k} B^{n-2k} \right\rangle_{\psi,\alpha,\gamma_2} = \frac{1}{8\pi^2} \int\limits_0^{2\pi} \mathrm{d}\gamma_2 \int\limits_0^{\pi} \mathrm{d}\alpha \, \sin\alpha \int\limits_0^{2\pi} \mathrm{d}\psi \, A^{2k} B^{n-2k} \tag{16}$$

The following auxiliary formulae can be further be applied; for averaging over an azimuth $\psi$,

$$\langle \sin^m\psi \, \cos^n\psi \rangle_\psi = \begin{cases} \frac{(m-1)!!(n-1)!!}{(m+n)!!} & \text{if } m \text{ and } n \text{ are even} \\ 0 & \text{otherwise} \end{cases} \tag{17}$$



and for $\gamma_2$, for averaging over a polar angle $\alpha$.

$$\langle \cos^n \alpha \, \sin^m \alpha \rangle_{\cos \alpha} = \frac{m!! \, (n-1)!!}{(m+n+1)!!} \quad \text{for even } m \text{ and } n \,. \tag{18}$$

Both relations can be proven by complete induction; see the Supplement S1. The operation !! denotes the double factorial $(n! = n!! \cdot (n-1)!!)$.

Insertion of the auxiliary formulae yields

$$\left\langle A^{2k} B^{n-2k} \right\rangle_{\text{or}} = \sum_{p=0}^{k} \sum_{q=0}^{k} \sum_{r=0}^{n-2k} \frac{k!}{p!q! \, (k-p-q)!} \binom{n-2k}{r} \left\langle a^p b^q c^{k-p-q} d^r e^{n-2k-r} \right\rangle$$

$$\times \frac{(p+r-1)!! \, (q+n-2k-r-1)!!}{(p+q+n-2k)!!} \cdot \frac{1+(-1)^{p+r}}{2} \cdot \frac{1+(-1)^{q+n-2k-r}}{2} \,. \tag{19}$$

We insert eqn. (19) into eqn. (13) and replace the trigonometric expressions by complex exponentials (again applying the
binomial law ):

$$\sin^{2k} \frac{\omega_r t}{2} \sin^{n-2k} \omega_r t = \frac{1}{(2i)^n} \sum_{a=0}^{2k} \sum_{b=0}^{n-2k} \binom{2k}{a} \binom{n-2k}{b} (-1)^{a+b} \, e^{i(n-k-a-2b)\omega_r t} \tag{20}$$

$\langle \Phi^n \rangle$ is described now by a very long expression which can be found in the Supplement S2.

Step 3 (assembling the FID and Fourier analysis):
After inserting $\langle \Phi^n \rangle$ into eqn. (8) and comparing with eqn. (9) we obtain for the Fourier coefficients

$$I_m = \sum_{n=m}^{\infty} \frac{n!}{2^n} \left( \frac{\omega_0 \, \delta}{\omega_r} \right)^n \sum_{k=0}^{[n/2]} \sum_{b=0}^{n-2k} \frac{(-1)^{n-k-m-b}}{(n-k-m-2b)!(3k-n+m+2b)!(n-2k-b)!b!}$$

$$\times \sum_{p=0}^{k} \sum_{q=0}^{k} \sum_{r=0}^{n-2k} \frac{k!}{p!q! \, (k-p-q)!} \binom{n-2k}{r} \frac{2^{4k+r+p}}{6^n} (-1)^r$$

$$\times \frac{(p+r-1)!! \, (q+n-2k-r-1)!!}{(p+q+n-2k)!!} \cdot \frac{1+(-1)^{r-p}}{2} \cdot \frac{1+(-1)^{q+n-2k-r}}{2}$$

$$\times \sum_{s=0}^{q} \sum_{t=0}^{(k-q-p)} \sum_{u=0}^{(n-2k-r)} (-1)^{q-s+u} \binom{q}{s} \binom{k-p-q}{t} \binom{n-2k-r}{u} \cdot \frac{(2k)!! \cdot (p+r+2s+2t+2u-1)!!}{(2k+p+r+2s+2t+2u+1)!!} \times$$

$$\times \sum_{v=0}^{(p+2s+2t+u)} \sum_{w=0}^{(n-2k-r-u)} (-1)^v 3^{n-2k+p-r+2s+2t-v-w} \binom{p+2s+2t+u}{v} \binom{n-2k-r-u}{w} \eta^{v+w+2k-p-2s-2t+r}$$

$$\times \frac{(v+w-1)!!(2k-p-2s-2t+r-1)!!}{(v+w+2k-p-2s-2t+r)!!} \cdot \frac{1+(-1)^{v+w}}{2} \cdot \frac{1+(-1)^{r-p}}{2} \tag{21}$$

This is our first core result. Within the infinite limits, this is an *exact* expression for the intensity of the SSB of $m$-th order, not an approximation. In practice, this equation can be used for generating terms of arbitrary order. Importantly, its numerical



evaluation will be appreciably faster than a numerical powder average of eqn. (8). However, it cannot be applied immediately because of its complex structure. However, using symbolic software it is easily possible to create polynomials for $I_m$ to reach arbitrary precision; a Mathematica notebook is given in the Supplement S3. Just for the purpose of illustration, we here provide the expression for the centerband up to 12th order in $\omega_0/\omega_r$ (abbreviations: $K_1 := 3 + \eta^2$ and $K_2 := 1 - \eta^2$):

$$I_0 = 1 - \frac{K_1^2}{20}\left(\frac{\delta\omega_0}{\omega_r}\right)^2 + \frac{227\,K_1^2}{181\,440}\left(\frac{\delta\omega_0}{\omega_r}\right)^4 - \frac{49\,471\,K_1^3 + 4\,428\,K_2^2}{2\,802\,159\,360}\left(\frac{\delta\omega_0}{\omega_r}\right)^6 + \frac{K_1\left(1\,466\,405\,K_1^3 - 709\,776\,K_2^2\right)}{9\,146\,248\,151\,040}\left(\frac{\delta\omega_0}{\omega_r}\right)^8$$
$$- \frac{K_1^2\left(286\,311\,167\,K_1^3 - 494\,915\,400\,K_2^2\right)}{281\,521\,518\,089\,011\,200}\left(\frac{\delta\omega_0}{\omega_r}\right)^{10}$$
$$+ \frac{998\,271\,153\,509\,K_1^6 - 2\,160\,K_2^2\left(1\,577\,931\,893\,K_1^3 + 218\,222\,883\,K_2^2\right)}{209\,789\,835\,279\,931\,146\,240\,000}\left(\frac{\delta\omega_0}{\omega_r}\right)^{12}$$
$$\tag{22}$$

Analogous formulae for all SSB up to fourth order are provided in the Supplement S4.

## 2.4 Properties of the polynomials

We consider it useful to discuss a few properties of the polynomials. Terms with even powers of $R$ are symmetric and thus invariant with respect to change of the sign of the sideband order, while the odd-power terms are skew-symmetric. This can be written as

$$I_{\pm m} = \sum_{n=0}^{\infty} b_{m;2n} R^{2n} \ \pm\ K_2 R \cdot \sum_{n=0}^{\infty} b_{m;2n+1} R^{2n}\ . \tag{23}$$

This property can be used for constructing polynomials which might possess better convergence behavior, namely sums and differences,

$$I_m + I_{-m} = 2 \sum_{n=0}^{\infty} b_{m;2n} R^{2n}$$

$$I_m - I_{-m} = 2\,K_2 R \cdot \sum_{n=0}^{\infty} b_{m;2n+1} R^{2n}\ . \tag{24}$$

The meaning of this substitution is that the properties of anisotropy and asymmetry can be separated almost completely into one of the combinations. A physical rationale is that the difference between the two SSB of first order is the larger, the more asymmetric the static powder pattern is, i.e. for $\eta = 1$, hence $K_2 = 0$ (symmetric static line!) we expect $I_m = I_{-m}$. The axially symmetric tensor ($\eta = 0$, i.e. $K_2$ maximum) yields the most asymmetric shape, so the difference between these two SSB should be large. Contrarily, the average (or sum) of both is expected to be the larger, the larger is the anisotropy $\delta$.

Specifically, to combine the SSB of first order, we define the quantities $I_+ = (I_1 + I_{-1})/I_0$ and $I_- = (I_1 - I_{-1})/I_0$. By polynomial division we obtain in decimal notation

$$I_+ = \ 0.044\,444\,44\,w^2 + 0.000\,723\,104\,0\,w^4 + \left(4.900\,449 \times 10^{-6} - 2.127\,502 \times 10^{-5} q^2\right)w^6$$
$$- \left(1.156\,835 \times 10^{-7} + 4.384\,854 \times 10^{-7} q^2\right)w^8 - \left(4.660\,243 \times 10^{-9} - 2.669\,296 \times 10^{-9} q^2\right)w^{10}$$
$$- \left[8.035\,180 \times 10^{-11} - \left(3.397\,989 \times 10^{-10} + 3.261\,380 \times 10^{-11} q^2\right)q^2\right]w^{12} \tag{25}$$





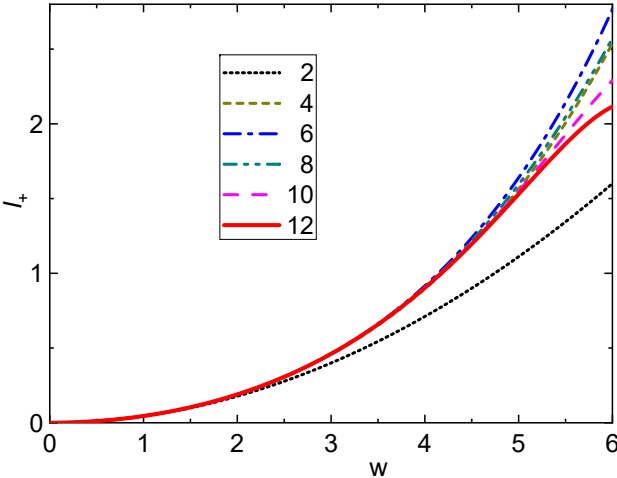

**Figure 2.** SSB combination $I_+$ *vs.* $w = (\omega_0 \delta / \omega_r) \sqrt{3 + \eta^2}$ for $q = 1$ and different degrees of approximation. The numbers in the legend are the maximum powers of $w$ considered.

and

$$
\begin{aligned}
I_- = &- 0.019\,047\,62\, q\, w^3 \left[ 1 + 0.020\,959\,6\, w^2 + 0.000\,181\,331\, w^4 \right. \\
&\left. - \left( 2.592\,966 \times 10^{-6} - 2.214\,666 \times 10^{-6} q^2 \right) w^6 - \left( 1.296\,641 \times 10^{-7} - 2.065\,980 \times 10^{-7} q^2 \right) w^8 + O[w]^{10} \right] ,
\end{aligned}
\tag{26}
$$

where the two independent variables have the meaning

$$
w := \frac{\omega_0 \delta}{\omega_r} \sqrt{K_1} = \frac{\omega_0 \delta}{\omega_r} \sqrt{3 + \eta^2} \quad \text{and} \quad q := \frac{K_2}{K_1^{3/2}} = \frac{1 - \eta^2}{(3 + \eta^2)^{3/2}} \ .
\tag{27}
$$

The variable $w$ represents the ratio between anisotropy and spinning speed together with an additional $\eta$-dependent factor. $q$ represents the tensor asymmetry in a way that $\eta = 0 \rightarrow q = 1/\sqrt{27}$ and $\eta = 1 \rightarrow q = 0$. In this representation, the asymmetry ($\eta$) dependence resides almost completely in the prefactor, while the terms in the rectangular brackets vary less than $I_+$ if the asymmetry varies between its extremes, to be addressed below.

We thus summarize the advantages of such an approach:

1. These combinations depend on the two dimensionless variables $w$ and $q$, which enables an easy extraction of the tensor parameters ($w$ expresses the ratio of anisotropy and spinning frequency including an $\eta$ component).

2. Fitting a ratio removes the need for fitting an additional, anyways arbitrary amplitude.

3. The powers of $R$ increase from term to term by 2 instead of 1 as in the case of single SSB, leading to less terms needed for a sufficiently good approximation. $I_+$ curves assuming $\eta = 1$ for different degrees of approximation are shown in





Fig.2, and demonstrate the relevance of higher-order terms for a given value of $w$. We can conclude that the use of the first two terms only ($w^4$) is a very good approximation up to $w \approx 4$, while terms up to 12th order are required to cover $w \leq 5$.

4. For not too small $\omega_r$, $w$ dominates in the coefficients over $q$. Particularly, up to the 5th power in $w$, the sums also of the other SSB depend only on $K_1$ and the differences depend linearly on $q$ (the prefactor), i.e. even for the maximum

value $w$ for which the 12th-order approximation is well justified ($w \approx 5$), we get $I_+ = 1.518$ for $\eta = 0$ and $I_+ = 1.530$ for $\eta = 1$, which makes a difference of only 0.8%.

Therefore, the two eqs. (24) provide a means to separate the dependencies on the two invariants. To stress this point, Fig. 3a shows $I_+$ *vs.* $w$ for a range $\eta$ values, normalized to its dependence for $\eta = 1$. The variation range, somewhat amplified by the narrow plotted interval, is less than 1%, thus very small, which confirms item 3. In contrast, $I_-$ features a strong variation with

$q$ as well as $\eta$. However, looking again at the normalized dependence of $I_-$ on $w$ (where we now need to distinguish positive

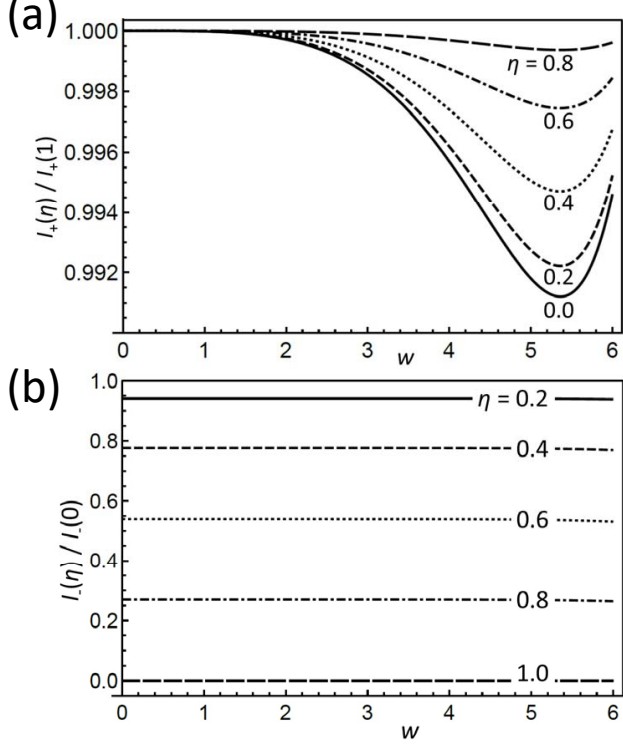

**Figure 3.** (a) $I_+$ and (b) $I_-$ for different $\eta$, normalized to their values at $\eta = 1$ and $\eta = 0$, respectively, as a function of $w$. Note the smallness of the differences between the $I_+$ curves in (a) of different $\eta$; they are less than 1%. Remarkably in (b) is the almost independent ratio between the $I_-$ of different $\eta$.



and negative values) plotted in Fig. 3b, we prove that this dependence is almost completely governed by the prefactor. In other words, $\eta$ is very sensitively encoded in $I_-$ once $w$ is rather precisely determined from $I_+$

In summary, two different approaches lend themselves to the analysis of actual data with the aim of extracting $\delta$ and $\eta$:

(1) One can include as many as possible SSB, trying to fit $K_1$ and $K_2$ by a fit to the SSB intensities including a normalization
factor as fit parameter. The tensor invariants are readily obtained by solving the given system of 2 nonlinear equations.

(2) One can consider only first-order sidebands, check the magnitude of $I_+$, then decide which level of approximation has to be used, then calculate $w$, then $q$, possibly iteratively until self-consistency.

## 2.5 Derivation of the 6th order polynomials for the 2D syncMAS sidebands

Step 1 (Phase):

For this treatment, two additional frames are needed as compared to the 1D MAS case of an isotropic sample, see Section 2.1.2. As outlined there, we assume uniaxial symmetry around the unique deformation axis (director). Note that the rotor should be packed in such a way that the director is perpendicular to the spinning axis, i.e. $\beta_2 = 90°$. We employ our tensor-based approach to performing the rotation transformations, see also Section 2.1.2. Taking $\delta_{\mathrm{L}}$ and $\delta_{\mathrm{M}}$ as the column vectors representing the
CST in the lab frame and in its main frame, respectively, we arrive at the following series of transformations:

$$\delta_{\mathrm{L}} = \mathbf{R_y}(\vartheta_{\mathrm{MA}}) \cdot \mathbf{R_z}(\gamma) \cdot \mathbf{R_y}\left(\frac{\pi}{2}\right) \cdot \mathbf{R_z}(\varphi) \cdot \mathbf{R_y}(\beta) \cdot \mathbf{R_z}(\varepsilon) \cdot \mathbf{R_y}(\alpha) \cdot \mathbf{R_z}(\psi) \cdot \delta_{\mathrm{M}} \tag{28}$$

The instantaneous frequency can now be calculated:

$$
\begin{aligned}
\omega = \omega_0\, \delta\, \tfrac{1}{3\sqrt{2}} \Big\{ &-2\sin 2\gamma \left( [\cos\beta\,(E_1\cos\varepsilon + E_2\sin\varepsilon) - \sin\beta\,(E_3\cos 2\varepsilon + E_4\sin 2\varepsilon)]\cos\varphi \right.\\
&+ \left[\cos 2\beta\,(E_2\cos\varepsilon - E_1\sin\varepsilon) + \tfrac{1}{2}\sin 2\beta\,(\sqrt{3}E_5 - E_4\cos 2\varepsilon + E_3\sin 2\varepsilon)\right]\sin\varphi \Big)\\
&+2\sqrt{2}\cos\gamma \left( [\cos 2\beta\,(E_2\cos\varepsilon - E_1\sin\varepsilon) + \tfrac{1}{2}\sin 2\beta\,(\sqrt{3}E_5 - E_4\cos 2\varepsilon + E_3\sin 2\varepsilon)]\cos\varphi \right.\\
&+ \left[-\cos\beta\,(E_1\cos\varepsilon + E_2\sin\varepsilon) + \sin\beta\,(E_3\cos 2\varepsilon + E_4\sin 2\varepsilon)\right]\sin\varphi \Big)\\
&-2\sqrt{2}\sin\gamma \left( [\sin\beta\,(E_1\cos\varepsilon + E_2\sin\varepsilon) + \cos\beta\,(E_3\cos 2\varepsilon + E_4\sin 2\varepsilon)]\cos 2\varphi \right.\\
&+ \tfrac{1}{4}\left[2\sin 2\beta\,(E_2\cos\varepsilon - E_1\sin\varepsilon) + (3+\cos 2\beta)(E_4\cos 2\varepsilon - E_3\sin 2\varepsilon) + 2\sqrt{3}E_5\sin^2\beta\right]\sin 2\varphi \Big)\\
&+\cos 2\gamma \left( \tfrac{1}{4}\left[(3+\cos 2\beta)(E_4\cos 2\varepsilon - E_3\sin 2\varepsilon) + 2\sin 2\beta\,(E_2\cos\varepsilon - E_1\sin\varepsilon) + 2\sqrt{3}E_5\sin^2\beta\right]\cos 2\varphi \right.\\
&- \left[\sin\beta\,(E_1\cos\varepsilon + E_2\sin\varepsilon) + \cos\beta\,(E_3\cos 2\varepsilon + E_4\sin 2\varepsilon)\right]\sin 2\varphi\\
&+ \tfrac{\sqrt{3}}{2}E_5\,(3\cos^2\beta - 1) + \tfrac{3}{2}\left[\sin 2\beta\,(-E_2\cos\varepsilon + E_1\sin\varepsilon) + \sin^2\beta\,(E_4\cos 2\varepsilon - E_3\sin 2\varepsilon)\right] \Big) \Big\}
\end{aligned}
\tag{29}
$$

We obtain the phase accumulated from the end of signal excitation ($t = 0$) to time $t$ by time integration:

$$\Phi = \frac{\delta \cdot \omega_0}{\omega_r}\left[(aD_1 + bD_2)\cos\varphi + (bD_1 - aD_2)\sin\varphi + (cD_3 + dD_4)\cos 2\varphi + (dD_3 - cD_4)\sin 2\varphi + \sqrt{3}\, f\, D_4\right] \tag{30}$$





This is again a sum of products of solely $t$-dependent and of solely orientation-dependent terms. The former are

$$D_1 = -\frac{\sqrt{2}}{3} \sin \omega_r t_2 \sin \left(\omega_r t_2 + 2\omega_r t_1 + 2\gamma_{00}\right)$$

$$D_2 = \frac{2}{3} \left[\sin \left(\omega_r t_2 + \omega_r t_1 + \gamma_{00}\right) - \sin(\omega_r t_1 + \gamma_{00})\right]$$

$$D_3 = \frac{2}{3} \left[\cos \left(\omega_r t_2 + \omega_r t_1 + \gamma_{00}\right) - \cos(\omega_r t_1 + \gamma_{00})\right]$$

$$D_4 = \frac{1}{\sqrt{6}} \sin \omega_r t_2 \cos \left(\omega_r t_2 + 2\omega_r t_1 + 2\gamma_{00}\right) \tag{31}$$

The latter can be separated further into factors which depend on the three angles $\varphi$, $\beta$ and $\varepsilon$ describing the orientation of the molecular segments:

$$a := \cos\beta \left(E_1 \cos\varepsilon + E_2 \sin\varepsilon\right) - \sin\beta \left(E_3 \cos 2\varepsilon + E_4 \sin 2\varepsilon\right)$$

$$b := \cos 2\beta \left(E_2 \cos\varepsilon - E_1 \sin\varepsilon\right) + \frac{1}{2}\sin 2\beta \left(\sqrt{3}E_5 - E_4 \cos 2\varepsilon + E_3 \sin 2\varepsilon\right)$$

$$c := \sin\beta \left(E_1 \cos\varepsilon + E_2 \sin\varepsilon\right) + \cos\beta \left(E_3 \cos 2\varepsilon + E_4 \sin 2\varepsilon\right)$$

$$d := \frac{1}{4}\left[2\sin 2\beta \left(E_2 \cos\varepsilon - E_1 \sin\varepsilon\right) + (3 + \cos 2\beta)\left(E_4 \cos 2\varepsilon - E_3 \sin 2\varepsilon\right) + 2\sqrt{3}E_5 \sin^2\beta\right]$$

$$f := \frac{\sqrt{3}}{2}E_5 \left(3\cos^2\beta - 1\right) + \frac{3}{2}\left[\sin 2\beta \left(-E_2 \cos\varepsilon + E_1 \sin\varepsilon\right) + \sin^2\beta \left(E_4 \cos 2\varepsilon - E_3 \sin 2\varepsilon\right)\right] \tag{32}$$

Here, the $E_i$ ($i \in \{1..5\}$) ("geometry factors") contain all information about the orientation of the CST PAF in the molecular frame:

$$E_1 = -\frac{\eta}{\sqrt{2}} \sin\alpha \, \sin 2\psi$$

$$E_2 = \frac{1}{2\sqrt{2}} \sin 2\alpha \left[3 - \eta \cos 2\psi\right]$$

$$E_3 = \frac{\eta}{\sqrt{2}} \cos\alpha \, \sin 2\psi$$

$$E_4 = \frac{1}{2\sqrt{2}} \left[\eta(1 + \cos^2\alpha)\cos 2\psi + 3\sin^2\alpha\right]$$

$$E_5 = \frac{\sqrt{3}}{2\sqrt{2}} \left[(3\cos^2\alpha - 1) + \eta\sin^2\alpha \cos 2\psi\right] \tag{33}$$

Step 2 (Orientational averaging, here up to $n = 6$)

Orientation averaging of the powers of $\Phi$ reads

$$\langle\Phi^n\rangle_{\text{or}} = \frac{1}{8\pi^2} \int\limits_0^{2\pi} \mathrm{d}\varphi \int\limits_0^\pi \cdot \, U\left(\beta\right) \, \sin\beta \, \mathrm{d}\beta \int\limits_0^{2\pi} \mathrm{d}\varepsilon \ \Phi^n(\varphi, \beta, \varepsilon) \,, \tag{34}$$

which now includes the non-isotropic, symmetric ODF (assuming equal probability of all $\varepsilon$). The $\beta$ dependence of $\Phi^n$ after $\varphi$ and $\varepsilon$ integration consists of $\cos^2\beta$ to powers $\leq 6$, which can be written as linear combinations of the Legendre polynomials $P_n(\cos\beta)$ with even $n \leq 12$. Hence, the result of the powder averaging procedure will be a linear combination of the orientational moments $\langle P_2\rangle$, $\langle P_4\rangle$, $\langle P_6\rangle$, $\langle P_8\rangle$, $\langle P_{10}\rangle$ and $\langle P_{12}\rangle$; see also eqn. (4). Due to the small values which are expected for



$\langle P_{10}\rangle$ and $\langle P_{12}\rangle$, the coefficients assigned to these orientational moments are neglected in the following. The phase powers contain powers and mixed products of the time-dependent terms $D_1..D_4$. Transforming again the trigonometric forms into exponential ones we obtain linear combinations of $e^{im\omega_r t_2}e^{ik\omega_r t_1}$.

Step 3 (assembling the FID and Fourier analysis):

Summing up the phase powers to obtain the FID expression corresponding to eqn. (8) and separating the $I_{mk}$ following eqn. (9) gives the complex 2D SSB intensities. All coefficients with odd $k$ actually vanish, which is the mathematical reflection of the fact that MAS rotation by $180°$ provides an invariant situation when the director is perpendicular to the spinning axis, i.e. $\beta_2 = 90°$. In other words, a full rotation replicates each arrangement twice if the director is perpendicular to the rotor axis (for other sample packing schemes, odd-order sidebands will appear, requiring somewhat more lengthy calculations).

Instead of the complex representation of the SSB intensities, we use the trigonometric representation because here the phase problems can be eliminated in a more efficient way. This will be discussed in detail in the application section. Fourier analysis along $t_1$ gives

$$I_m(t_1) = \sum_{k=0}^{4}\left[C_{m;2k}\cos\left(2k\omega_r t_1 + \gamma_{00}\right) + S_{m;2k}\sin\left(2k\omega_r t_1 + \gamma_{00}\right)\right] . \qquad (35)$$

The trigonometric SSB intensities $C_{mk}$ and $S_{mk}$ can be transformed to the complex ones and vice versa as follows:

$$\begin{aligned}
I_{m;2k} + I_{m;-2k} &= C_{m;2k}(k>0) \\
I_{m0} &= C_{m0} \Rightarrow I_{m;\pm 2k} = \frac{1}{2}\left(C_{m;2k} \mp iS_{m;2k}\right)(k>0) \\
iI_{m;2k} - iI_{m;-2k} &= S_{m;2k}
\end{aligned} \qquad (36)$$

Both kinds of coefficients contain the same information, however, the trigonometric coefficients include less terms. This arises from the $\pm$ sign which leads to a cancellation of some terms upon addition. This could have the advantage that the intervals between powers are larger and the error in neglecting higher powers above a certain value might be reduced.

The linear dependence of the SSB intensities on the orientational moments mentioned above has the consequence that also $C_{mk}$ and $S_{mk}$ are linear in the orientational moments:

$$C_{mk} = \sum_p C_{m,k,2p}\langle P_{2p}\rangle \quad ; \quad S_{mk} = \sum_p S_{m,k,2p}\langle P_{2p}\rangle \qquad (37)$$

$C_{m,k,2p}$ and $S_{m,k,2p}$ depend on $\delta$, $\eta$ and the two angles $\alpha$ and $\psi$ (which are the spherical coordinates of the molecular vector in the CST PAF, see above). Analytical expressions for them are listed for some low $m$ and $k$ in Supplement S5.





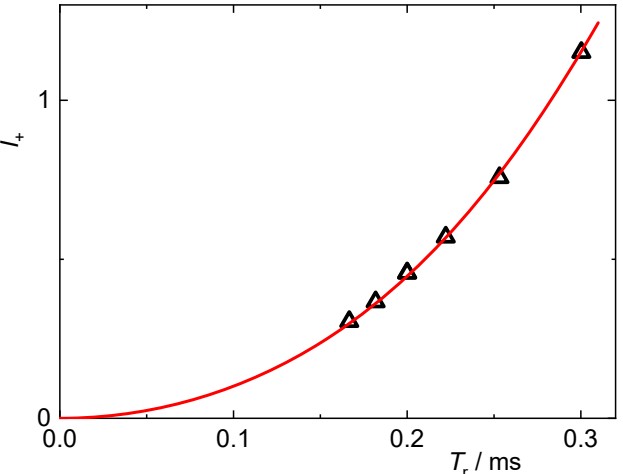

**Figure 4.** $I_+$ *vs.* rotation period $T_r$ for the COO resonance of glycine.

## 3 Applications of the polynomial approach

### 3.1 Tensor parameters from MAS SSB

Next, we describe a practical test of the polynomial expansion of SSB intensities to extract actual tensor parameters, and illustrate an effective procedure to reach this aim. We use glycine as an example and focus on the normalized sum and difference of the first-order SSB intensities relative to that of the centerband, $I_+$ and $I_-$, respectively. Fig. 4 shows the variation of $I_+$ with the rotor period $T_r$, i.e., the inverse of the variable spinning frequency. The largest value amounts to about 1.2. As shown in Figs. 2, this corresponds to $w < 4.5$ hence the 12th order approximation can be applied without compromise in accuracy from the theory side. With the given Larmor frequency of 100.6 MHz we obtain $\delta^2(3+\eta^2) = (2.165 \pm 0.01) \times 10^{-8}$.

The corresponding $I_-$ values give $q = (-6.1 \pm 2) \times 10^{-3}$. The apparently large relative uncertainty should be judged in proportion to the possible range $0 \le q \le 1/27 = 37 \times 10^{-3}$. The negative sign of $q$ means that $\delta$ is negative, i.e. that eigenvalue having the largest deviation from $\delta_{iso}$ is at lower CS (upfield-shifted, towards the right end of the spectrum).

Results for both carbon resonances of glycine are compiled in Table 1. For COO the agreement with the literature values is very satisfactory, in particular for $\eta$. The values for CH$_2$ deviate more an a relative scale; possible reasons are (i) that the spinning speed was optimized for investigation of COO, leading to small SSB intensities for CH$_2$ with its much narrower tensor, and (ii) that the dipolar coupling to $^{14}$N is not completely averaged by MAS because of the quadrupolar interaction, the contribution of which could be separated in the static single-crystal experiments of Griffin et al. (1975). The cross-polarization time was 1 ms which appears sufficiently to avoid some bias of the isotropic powder average, especially in a proton-rich enviroment.





**Table 1.** CS tensor parameters of glycine obtained by the polynomial procedure and comparison with literature values (Griffin et al., 1975; Haberkorn et al., 1981)

| Group | $\delta$/ppm | $\eta$ |
|---|---|---|
| COO measured | $-74.1 \pm 0.8$ | $0.98 \pm 0.02$ |
| literature value | $-70.65 \pm 1$ | $0.97$ |
| $CH_2$ measured | $23.46 \pm 0.25$ | $0.60 \pm 0.03$ |
| literature value | $20 \pm 1$ | $0.94$ |

### 3.2 Segmental orientation from syncMAS

We now turn to a demonstration of our approach in analyzing syncMAS data to extract molecular orientations in a uniaxially oriented sample. We first address the chosen sample and the polymer-physical background shortly and present $^{13}$C spectra together with the signal assignment in Section 3.2.1, and address the CST parameters in Section 3.2.2. The actual processing and analyses of 2D syncMAS data are covered in the following sections, where Section 3.2.3 addresses phase distortions in the 2D experiment, Section 3.2.4 summarizes the result of Fourier analysis in the indirect dimension, Section 3.2.5 discusses the

ambiguities related to PAF *vs.* molecular-frame orientations, and Section 3.2.6 finally provides the orientational moments and a discusssion.

### 3.2.1 Background and $^{13}$C CP MAS spectra

To illustrate the use of our approach to estimate orientational order in a practically relevant case, we turn to a polycarbonate (PC) sample (Makrolon GP clear 099 from Bayer) which was stretched in the glassy state to an elongation factor of 1.45. In

this process, the segments of the chains are expected to align. In an early application of 2D syncMAS to a similar sample polymer (Vogt et al., 1990), an order parameter $\langle P_2 \rangle$ of about 0.15 was reported for the methyl resonance at elongation. In more detailed work focusing on static 2D experiments of $^{13}$C-labeled PC (Utz et al., 1999), a different deformation geometry and a different angle convention for the director reference frame was used, rendering a comparison not straightforward. Here, we do not elaborate on the polymer physics details and merely use this sample for a proof of principle of the method.

In the earlier work (Vogt et al., 1990), only results for the methyl resonance were reported, and no compeding reason was given with regards to why the other resonances were not evaluated. This was probably due to the limited spinning speed and a corresponding lack of resolution. Moreover, the data analysis approach employing a fit using precalculated subspectra to extract the orientation degree (Harbison et al., 1987) implies fixed assumptions on the relation of the CST PAF and the director frame, which were probably not available for the other resonances. One key advantage of our approach is its flexibility to

change the CSA principle values and the related angles at no additional expense in calculation efficiency.



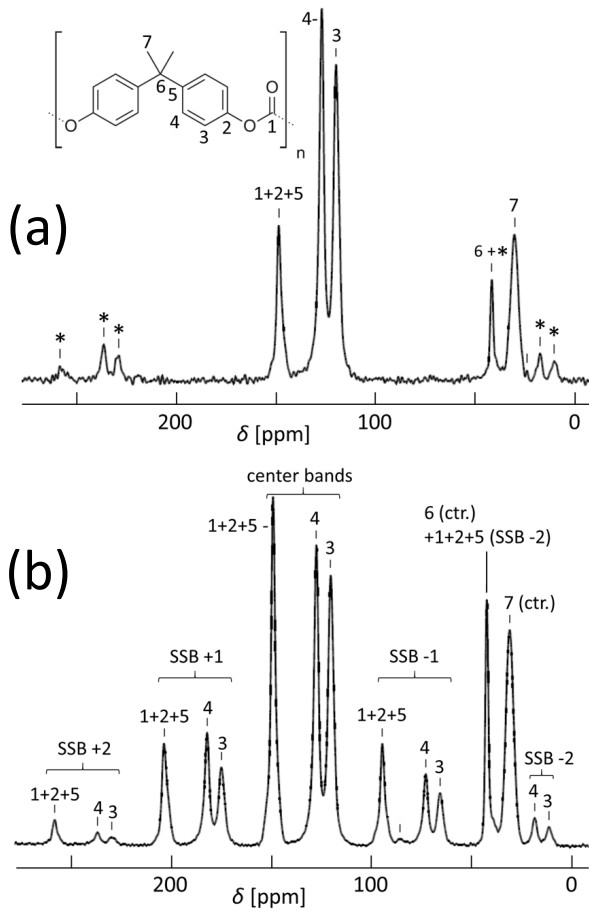

**Figure 5.** $^{13}$C CP MAS spectra of an isotropic PC sample with (a) 11 kHz and (b) 5.5 kHz spinning; for the resonance assignment see Williams et al. (1977). SSBs are marked by asterisks. The spectrum in (b) was taken with 20,000 scans.

As to experimental details (Djukic et al., 2020), we carefully avoided orientation effects from the injection molding procedure of the specimen via machining off the surface layer, and using precise video control of the true strain. A cylindrical piece of 3 mm outer diameter was cut with a dissecting knife from the center of the stretched specimen and inserted into a 4 mm MAS rotor such that the stretching direction was placed along the radius of the rotor. For the determination of the CSA principal values, an unstretched sample was powdered in order to fully remove possible anisotropy from the molding process, and compressed into a rotor. The cross-polarization (CP) MAS spectrum at 11 kHz spinning is shown in Fig. 5a.

For syncMAS, resolved spectral lines with unique assignments are needed. This is the case for the aromatic CH groups (C3 and C4 in Fig. 5a), but the carbonate (CO$_3$, C1) and both quaternary aromatic carbons (C$_q$, C2 and C5) are almost fully superimposed. Here only an approximate analysis of the combined signal is possible. As to the methyl group (CH$_3$, C7), we






**Table 2.** CST parameters of all $^{13}$C positions in PC. The data for C1 and C2 are from Robyr et al. (1998). Those for C3, C4, C5 and C7 were obtained by the polynomial procedure. For C5, a decomposition procedure was applied (see text). Column C5' contains CST data of C5 but obtained by mutual exchange of $x_1$ and $x_3$ axes. In all cases the eigenvalues are calculated from $\delta_{\mathrm{iso}}$, $\delta$ and $\eta$.

| Parameter | C1 | C2 | C3 | C4 | C5 | C5' | C7 |
|---|---|---|---|---|---|---|---|
| $\delta_{\mathrm{iso}}$ | 147.1 | 148.3 | 120.5 | 127.7 | 149.3±1.5 | 149.3±1.5 | 31±2 |
| $\delta$/ppm | 89 | 92 | -93.7±1 | $-103.8\pm1$ | -106.4±3.6 | 78.5±1.4 | $-34\pm5$ |
| $\eta$ | 0.39 | 0.54 | 0.367±0.04 | 0.407±0.025 | 0.47±0.08 | 1.73±1.5 | 0.5±0.2 |
| $\delta_{11}$/ppm | 84.9 | 77.9 | 184.5±2 | 200.6±1.5 | 227.5±3.5 | 42.9 | 56.5±4.5 |
| $\delta_{22}$/ppm | 120.1 | 127.2 | 150.2±1.9 | 158.4±1.3 | 177.5±3.5 | 177.5 | 39.5±4.5 |
| $\delta_{33}$/ppm | 236.4 | 239.9 | 26.8±1 | 23.8±1 | 42.9±4.5 | 227.5 | -3±5.5 |

have sufficient SSB intensity only when the spinning speed is rather low, then leading to potential overlap with the more numerous aromatic SSBs. This can be seen even in Fig. 5 of Vogt et al. (1990), which shows PC spectrum at a spinning speed which is sufficient to have both CH$_3$ and C$_q$ SSB, however, the separation between the SSB seems insufficient for truly precise analysis. In our hands, a spinning frequency of 5.5 kHz was the best compromise for a joint analysis, see Fig. 5b.

### 3.2.2    Tensor parameters in PC

The availability of precise CST components and their orientation of the resonances of PC is not optimal; only for the CO$_3$ and one of the C$_q$ (C2) data are available (Robyr et al., 1998; Utz et al., 1999). In Table 2 we summarize all CST data which were used for the calculation of orientational moments below.

     The orientation of the PAF in the molecular frame cannot be deduced from the methods discussed in this work. We assume that the eigenvalues follow the usual assignment for aromatic carbons, i.e., the axis related to largest eigenvalue (lowest shield-
ing) is parallel to the C–H bond, the axis related to the intermediate eigenvalue is perpendicular to C–H and in the ring plane, and the axis related to the lowest eigenvalue (largest shielding) is perpendicular to the ring plane. Possible deviations from these orientations are commonly reported to be in the range of a few degrees only and are thus neglected. The numbering of the axes, however, depends on the sign of $\delta$ corresponding to the convention mentioned above: For $\delta < 0$, $x_3$ is related to the lowest eigenvalue and $x_1$ is related to the largest eigenvalue, i.e. $x_3 \perp$ ring and $x_1 \parallel$ C–H. For $\delta > 0$, these axes have to be
exchanged, see Fig. 6.

     Unfortunately, the resonances of C1, C2, and C5 are not resolved but rather almost fully overlapped. These positions, however, play an important role in the data analysis of the syncMAS experiments. If the CST of all three positions were known, the syncMAS signals could be calculated as superposition of the three curves. For both C1 and C2, the CST eigenvalues are known from experiments on selectively $^{13}$C-labeled samples: $\delta = 89$ ppm, $\eta = 0.39$ for C1 and $\delta = 92$ ppm, $\eta = 0.54$ for C2
(Utz et al., 1999). Hence, the missing CST of C5 (one of the C$_q$) can be estimated if the SSB pattern of the line at 150 ppm is





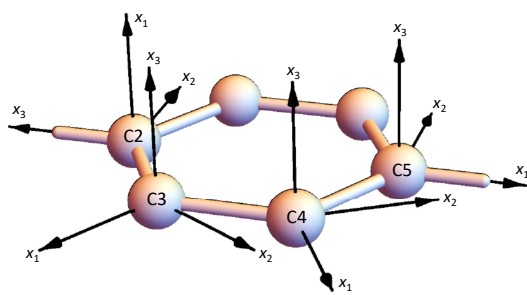

**Figure 6.** Orientation of the $^{13}$C CST PAFs in the benzene ring (C3,4 are C–H and C2,5 are the *para*-substituted $C_q$). The orientation of the C5 PAF corresponds to the convention mentioned above; for the alternative orientation, $x_1$ axis and $x_3$ axis had to be exchanged.

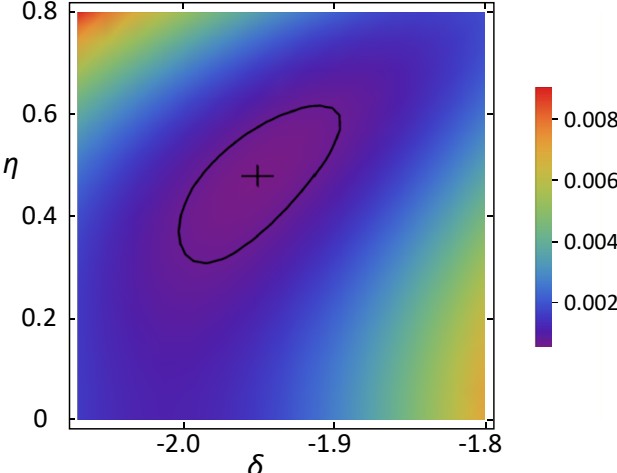

**Figure 7.** $\chi^2$ *vs.* anisotropy and asymmetry parameters of C5 for $p = 0.5$. The cross marks the minimum position; the solid line shows the confidence region.

considered as linear combination of the SSB patterns of one C1, two C2 and two C5 per monomer unit:

$$I_m = \frac{p\, I_m^{C1} + 2\, I_m^{C2} + 2\, I_m^{C5}}{p + 4} \tag{38}$$

This equation permits the calculation of the SSB pattern of C5, where $p$ describes the relative CP efficiency of $^{13}$CO$_3$, which is expected to be lower than that of C2 and C5 because of a larger distance of C1 to any protons as compared with C2 and C5 (thus, $0 < p \leq 1$). The summed square deviation $\chi^2$ between measured and best-fit SSB intensities has a minimum at $p \approx 0.5$. Fig. 7 shows a 2D $\chi^2$ map *vs.* anisotropy and asymmetry parameter using the so-estimated $p = 0.5$. With the polynomial method we obtained for carbon position C5: $\delta = (-106.6 \pm 4)$ ppm and $\eta = 0.48 \pm 0.08$. These values depend only weakly on $p$; the error intervals include this already.


Returning to the PAF orientations, see again Fig. 6, $x_3$ denotes the most-shielded direction (along the ring normal) and $x_1$ the least-shielded one (along the C–H bond). This numbering is in agreement with the definitions from Section 2.1.1 as long as






the anisotropy is positive. In the case of C5, however, $\delta < 0$. This means that the most-shielded direction is now that direction which belongs to that CST eigenvalue which deviates mostly from $\delta_{\mathrm{iso}}$. Following the definition from above, $x_1$ and $x_3$ had to exchange their directions. For the evaluation of the syncMAS data, however, it would be advantageous if both C2 and C5 were placed is a common frame. Then, for roughly 4/5 of the intensity ot this valuable signal, we have reliable CST values

and orientations, and small uncertainties related to the $CO_3$ resonance will not matter much. If the PAF of C2 is used also for C5, anisotropy and asymmetry parameter of C5 change to $\delta = (78.2 \pm 1)$ ppm and $\eta = 1.71 \pm 0.15$. We use these values in the following. The unusual value of $\eta$ is a consequence of the exchange of axes. One can of course easily check that the CST invariants as well as the eigenvalues are not influenced.

### 3.2.3   Fourier transform and 2D phase distortion

The relevant practical problem that is only partially described in the leading reference (Harbison et al., 1987) is the linear phase distortion along the indirect frequency dimension $\omega_1$ arising from the unknown angle $\gamma_{00}$ between the sample director and the rotor position that triggers the start of the pulse sequence in the case of $t_1 = 0$. This phase distortion superimposes with the "normal" phase shifts arising from quadrature detection and the pre-acquisition delay. Utmost stability of the spectrometer over the long-lasting experiment is required to resolved the related issues. This concerns in particular the signal excitation

(CP conditions). It is thus advisable to run a series of identical 2D syncMAS spectra and sum them up to reduce the effects of spectrometer drift along $t_1$. In addition, one can check the stability *via* the following combination of SSB which is almost independent of $t_1$:

$$0.8819 \cdot I_2 + 1.0121 \cdot I_1 + I_0 + 0.5609 \cdot I_{-1} + 0.8761 \cdot I_{-2} \tag{39}$$

This quantity should be constant within 0.1% across the different $t_1$ increments.

Referring to eqn. (35), the SSB intensities oscillate with $t_1$. This is valid for the real as well as for the imaginary parts of the spectra. For the special case of the director being perpendicular to the rotor axis, the 2D FID can be written as

$$FID\left(t_1, t_2\right) = \sum_{m,k} I_{mk} \exp\left[2k\omega_r\left(t_1 - t_0\right)\right] \cdot \exp\left[i(m\omega_r t_2) + \varphi\left(\omega_2\right] \cdot R\left(t_2\right) , \tag{40}$$

where $\varphi\left(\omega_2\right)$ is the phase distortion (constant, linear, ...) in the direct dimension, $R\left(t_2\right)$ denotes signal damping during acquisition and $t_0$ is the unknown delay corresponding to $\gamma_{00}$. The term $-2k\omega_r t_0$ corresponds to a linear phase distortion along $\omega_1$.

Both distortions sum up to a total phase distortion of $\varphi\left(\omega_2\right) - 2k\omega_r t_0$. They were separated *via* a procedure described in the following.

Performing only an FT along direct dimension, we obtain spectra $Sp\left(t_1, \omega_2\right)$ with absorptive ($A$) and dispersive ($D$) spectral components of the centerband and the SSBs, whose overall intensities oscillate along indirect dimension:

$$\begin{aligned} Sp\left(t_1, \omega_2\right) = e^{i\varphi(\omega_2)} \sum_k \{ & A \cdot C_{mk} \cos 2k\omega_r\left(t_1 - t_0\right) - D \cdot S_{mk} \sin 2k\omega_r\left(t_1 - t_0\right) \\ & + i\left[D \cdot C_{mk} \cos 2k\omega_r\left(t_1 - t_0\right) + A \cdot S_{mk} \sin 2k\omega_r\left(t_1 - t_0\right)\right] \} \end{aligned} \tag{41}$$



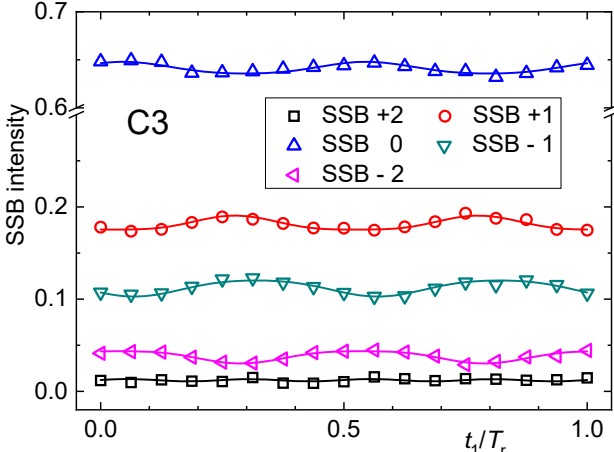

**Figure 8.** SSB oscillations along $t_1$ of carbon C3.

This means that for a general value of $t_1$ an apparent phase distortion is detected because of an inevitable mixing of sin and cos terms already without $\varphi(\omega_2)$. Because of variations of the $C_{mk}$ and $S_{mk}$, we will thus have different phase distortions already from signal to signal within each 1D spectrum! An exception is $t_1 - t_0 = n \cdot T_r/2$, because here the sin terms vanish.

A simple if not simplistic and laborious solution is to phase each signal of each slice separately and thus evaluate its intensity as a function of $t_1$. This procedure is only possible for sufficiently resolved spectral peaks and when neighboring peaks do not differ strongly in their phase. Results of such analyses are exemplarily shown in Fig. 8. The oscillations can then be fitted with a combination of sine and cosine dependencies to determine the prefactors of the different harmonics, but a close look at eqn. (41) reveals that an extraction of the $C_{mk}$ and $S_{mk}$ is nearly impossible, because the absorptive and dispersive components of the spectra along $\omega_2$ have been mixed. Therefore, the shown oscillations merely give a qualitative impression of the orientation degree in the sample, and a quantitative analysis is possibly only *via* a brute-force numerical approach.

A separation of both phase contributions is possible, provided that the $t_1$ incrementation is equidistant with $N$ values over one rotation period $T_r$ ($t_1 = iT_r/2^n$ with $i = 0 \ldots 2^n$; $2^n$=16 in our case). Upon summation of all spectra, only terms without trigonometric functions survive:

$$G_0 \left( \omega_2 \right) = \sum_{n=0}^{N-1} Sp \left( \frac{nT_r}{N}, \omega_2 \right) = e^{i\varphi(\omega_2)} \left( A + i\, D \right) C_{m0} . \tag{42}$$

Eqn. (42) represents a 1D spectrum that is distorted solely by phase shifts along the direct dimension, hence, the appropriate parameters needed for phase correction can be determined on this basis only. This spectrum is shown as the bottom trace of Fig. 9. Subsequently all individual spectra obtained by the first FT can be corrected by these parameters, and phase distortions left in the spectra are only those arising from $t_0$. It is important to stress that this sum is not identical to the spectrum of the isotropic sample. Instead, it also depends on the orientational moments, see the corresponding equations in Supplement S5. The




reason is that the summation is an azimuthal average over the rotor positions upon signal excitation; for an isotropic powder

435  average, some sample orientations would be needed which cannot be reached by this uniaxial sample rotation.

### 3.2.4  Obtaining $C_{mk}$ and $S_{mk}$ by Fourier analysis in the indirect dimension

Conventional 2D FT is possible after applying the phase correction from the sum spectrum to all $\omega_2$ slices along $t_1$ (Harbison et al., 1987). But then only two slices will be close to having only absorptive lineshapes, one of which can be taken as the $t_1 = 0$ slice via a roll-over of the time axis. However, identifying this one may be ambiguous with limited data quality, and one may not

440  have a spectrum at exactly that condition. Only then would a purely first-order (frequency-dependent) phase correction along $\omega_1$ provide absorptive spectral lines. Still then, an additional ambiguity with regards to the sign of the higher-order sidebands arises, requiring the testing of different possibilities. In our hands, an alternative approach proved more feasible.

We suggest performing Fourier analysis separately for the real and imaginary parts of the result of the first FT along $t_2$. Under the given conditions, only even-numbered Fourier coefficients do not vanish. The $\omega_2$-dependent Fourier coefficients of

445  order $2k$ ($k \in \mathbb{N}$) of the intermediate spectra are

$$
\begin{aligned}
G_{\mathrm{re},2\mathrm{k}}(\omega_2) &= \tfrac{1}{\pi} \int_0^{2\pi} \mathrm{Re}\{\mathrm{S}(\mathrm{t}_1,\omega_2)\} \exp\{2\mathrm{i}k\omega_\mathrm{r}\mathrm{t}_1\} \, \mathrm{d}(\omega_\mathrm{r}\mathrm{t}_1) = [A(\omega_2)C_{mk} - i\, D(\omega_2)S_{mk}]\, e^{2ik\omega_\mathrm{r}t_0} \\
G_{\mathrm{im},2\mathrm{k}}(\omega_2) &= \tfrac{1}{\pi} \int_0^{2\pi} \mathrm{Im}\{\mathrm{S}(\mathrm{t}_1,\omega_2)\} \exp\{2\mathrm{i}k\omega_\mathrm{r}\mathrm{t}_1\} \, \mathrm{d}(\omega_\mathrm{r}\mathrm{t}_1) = [D(\omega_2)C_{mk} + i\, A(\omega_2)S_{mk}]\, e^{2ik\omega_\mathrm{r}t_0} \, .
\end{aligned}
\tag{43}
$$

The new set of spectra $G_{\mathrm{re;im},2\mathrm{k}}(\omega_2)$ can be phased now; the correction angle of $-2k\omega_\mathrm{r}t_0$ for $G_{\mathrm{re},2\mathrm{k}}(\omega_2)$ is constant with respect to $\omega_2$, so one can apply the usual criterion of an as-absorptive-as-possible spectrum. The real part of a corrected spectrum contains the $C_{mk}$ as prefactors of $A(\omega_2)$. Phasing of $G_{\mathrm{im},2\mathrm{k}}(\omega_2)$ by the same angle yields an absorptive signal in the

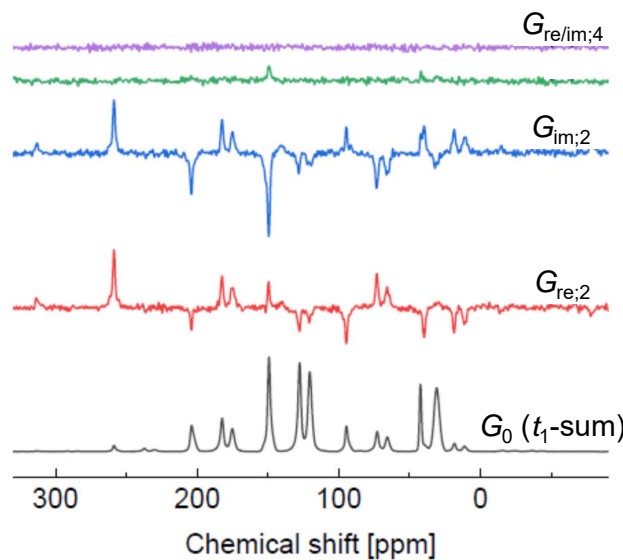

**Figure 9.** Spectra along $\omega_2$ (in ppm) encoding the $t_1$ Fourier coefficients as indicated. The lowest spectrum ist the sum of all slices in $t_1$.





imaginary part with $S_{mk}$ as prefactors. This is demonstrated for the higher Fourier coefficient spectra for $k = 0, 2$ and $4$ shown in Fig. 9. As already noted by Harbison et al. (1987), appearance of significant intensities in the spectra of order $k$ implies the relevance of orientational moments of similar order.

The peak intensities (integrals, heights) in these spectra can be identified with our exact solution for the sideband intensities, specifically, they can be used to estimate the $C_{mk}$ and $S_{mk}$:

$$\frac{C_{mk}}{C_{00}} = \frac{\mathrm{Re}\{\mathrm{G}_{\mathrm{re},2k}(\delta_{\mathrm{iso}} + \mathrm{m}\omega_{\mathrm{r}})\}}{\mathrm{Re}\{\mathrm{G}_0(\delta_{\mathrm{iso}})\}} \quad ; \quad \frac{S_{mk}}{C_{00}} = \frac{\mathrm{Im}\{\mathrm{G}_{\mathrm{im},2k}(\delta_{\mathrm{iso}} + \mathrm{m}\omega_{\mathrm{r}})\}}{\mathrm{Re}\{\mathrm{G}_0(\delta_{\mathrm{iso}})\}} \ . \tag{44}$$

The best values are those for $2k = 2$ ($C_{m0}$ have better S/N, but for the estimation of the orientational moments their difference to the isotropic SSB intensities have to be used, which are rather small). The possibility to extract higher orientational moments from the $C_{mk}$ and $S_{mk}$ by fitting depends on their accuracy and the availability of higher sideband orders. The full set of Fourier coefficients extracted from our syncMAS experiment on oriented PC is provided in Supplement S6.

### 3.2.5   Segment vector and CS tensor orientation

The theoretical considerations above are based on the assumption that all structural elements and therefore also all CST PAFs have a uniaxial distribution around the axis of the director frame; we used an isotropic average for the angle $\varepsilon$. Therefore, we have to be careful with regards to its definition on the segmental (monomer) level. Vogt et al. (1990) define the direction of the segment vector as being perpendicular to the $H_3C$-$C$-$CH_3$ plane. This will be used here only for the description of the orientation of this moiety, i.e., for the analysis of the results measured for the $CH_3$ resonance. For the other parts of the monomer this is of little benefit, because the intramolecular angles between this direction and other bonds are not known with sufficient accuracy.

Instead, for the other groups we use the connection line of the two ester oxygens of the carbonate group (C1). For the chemical environment of this position, reliable structural data were published by Utz et al. (1999). These authors indeed detected a distribution of tilt and torsion angles, so we used the averages for our purpose. These agree well with the results of SAXS experiments on crystalline diphenyl carbonate (King and Bryant, 1993). We considered also the latter results because they can be assumed to deviate only little from PC in the vicinity of the $CO_3$ group. From the data of both papers we estimated values for the relevant bond angles. For our data evaluation we used following values. The ring is tilted by $17.6°$ against the segmental vector (defining the "ring long axis", i.e., the connection line of the *para*-substituted carbons); the torsional angle around this axis is $53.2°$.

### 3.2.6   Orientational moments

Following eqn. (37), the multilinear dependence of the oscillation coefficients on the orientational moments is used for a multilinear regression procedure. All used experimental data are collected in the vector $\mathbf{Y}$, the orientational moments $\langle P_2 \rangle, \langle P_4 \rangle, \langle P_6 \rangle, \langle P_8 \rangle$ form the vector $\mathbf{P}$ and the coefficients $C_{m,k,2p}$ and $S_{m,k,2p}$ are elements of a matrix $\mathbf{X}$ with as many columns as orientational moments included, and rows determined by the available data. Eqn. (37) then reads

$$\mathbf{Y} = \mathbf{X}.\mathbf{P} \ . \tag{45}$$





**Table 3.** Orientational moments of PC stretched by a factor of 1.45 as obtained from our analyses and indicators of fitting quality.

| Position | $\langle P_2 \rangle$ | $\langle P_4 \rangle$ | $10\Delta I^2$ | $\chi^2$ |
|---|---|---|---|---|
| C1 + C2 + C5 | 0.234±0.04 | -0.187±0.12 | $1.0\times10^{-5}$ | $4.5\times10^{-5}$ |
| C3 | 0.21±0.05 | – | $1.2\times10^{-5}$ | $8.9\times10^{-5}$ |
| C4 | 0.27±0.1 | – | $1.2\times10^{-5}$ | $11.1\times10^{-5}$ |
| C7: $\psi$=0 | 0.22±0.08 | -0.76±0.4 | $1.8\times10^{-5}$ | $2.5\times10^{-5}$ |
| C7: $\psi$=90° | 0.100±0.024 | -0.065±0.01 | $1.8\times10^{-5}$ | $1.4\times10^{-5}$ |

The target quantity for optimization, i.e., the minimized sum square deviation ($\chi^2$) is given by

$$\mathbf{P}_{\mathrm{min}} := \underset{\mathbf{P}\in\mathbb{R}^N}{\mathrm{argmin}}\,[\mathbf{Y}-\mathbf{X}.\mathbf{P}]^2 = \left[\mathbf{X}^{\mathrm{T}}.\mathbf{X}\right]^{-1}.\mathbf{X}^{\mathrm{T}}.\mathbf{Y}\,. \tag{46}$$

In order to avoid the situation that a good fit is achieved by a too large number of physically irrelevant fitting parameters, we
proceeded as follows. In the first step, only $\langle P_2 \rangle$ was used. If the variance of this result $\chi^2 = [\mathbf{Y}-\mathbf{X}.\mathbf{P}]^2$ was exceeding the sum of squared experimental uncertainties $(\Delta\mathbf{Y})^2$, $\langle P_4 \rangle$ was added to the result vector $\mathbf{P}$, and so on. The results can be found in Table 3. The confidence intervals are determined as the variation of $\mathbf{P}$ which doubles the variance. The second to last column in Table 3 is the noise-related sum squared uncertainty of 10 SSB intensities considered. Only for the C1+C2+C5 combined signal does $\chi^2$ from the best fit exceed this value significantly. This may be an indication of our incomplete knowledge on the
geometric parameters of the involved resonances.

The $I_{mk}$ or alternatively the $C_{mk}$ and $S_{mk}$ suffer from the ambiguity that the addition of $T_{\mathrm{r}}/2$ to $t_0$ and sign inversion of all $I_{mk}$, $S_{mk}$ and $C_{mk}$ with odd $k$ lead to the same FID. This is related to the unknown linear phase correction along $\omega_1$. Within this experiment, there is no possibility to distinguish between the two situations. This means that one has to do two final fits, one with all $C_{mk}$ and $S_{mk}$ inverted for odd $k$. If one of these two cases leads to a physically implausible result, then this can
be used to identify the incorrect alternative. We have chosen the possibility which yields a positive orientational moment $\langle P_2 \rangle$ for the C1+C2+C5 combined signal.

The results for the protonated carbons have a somewhat larger uncertainty. This might arise from their sensitivity to small changes of the angle of rotation of rings around their long axis. Even though the angle was varied during data evaluation, $\chi^2$ remained at a level which is appreciably higher than the noise-related uncertainty. Moreover, $\langle P_4 \rangle$ variations lead to a rather
small increase of $\chi^2$; hence, these values are not shown in Table 3.

We observe a rather gratifying correspondence of the results obtained for the aromatic resonances, including the ones that overlap with the C1 (the CO$_3$ group). For an interpretation of these results, we remind that these orientation degrees correspond to a hypothetical segmental long axis, with respect to which we have positioned the CST PAFs (see the preceding subsection). This axis is defined to be the connection vector of the ester oxygens of the CO$_3$ group. Our values for are $\langle P_2 \rangle$ on averge even
somewhat higher than the value of about 0.15 published by Vogt et al. (1990), but a detailed comparison is difficult because





of the potential methodological issues of this work (see also below), and because of the different director frames used. In all, we note that our result is of the same order of magnitude, thus providing good confirmation of our efficient and (we hope) transparent approach to the data analysis. A notable and robust result is the comparably large and negative value of $\langle P_4 \rangle$ for two of our resonances. Our approach thus allows us to extract more information than achieved previously.

The state of the art concerning orientation effects in strained glassy PC was presented by Utz et al. (1999), as they have extracted the full ODF expanded in terms of up to 20 expansion coefficients from dedicated $^{13}$C static 2D experiments combined with isotope labeling. As already noted, a direct comparison with their results is difficult because of the different deformation geometry and the different angle conventions (essentially a rotation by 90°). Details will be deferred to a future publication. We can merely note that if we just consider a factor of $-2$ applied to their data to account for the 90° rotation of the reference

frame, their results for the second- and fourth-order expansion coefficients are of the same sign and magnitude as our $\langle P_2 \rangle$ and $\langle P_4 \rangle$. As shown in their ODF, the negative $\langle P_4 \rangle$ may be related to a population of main-chain segments oriented almost perpendicular to the stretching axis. Further systematic studies, enabled by our more efficient approach applicable to non-labeled samples, are planned.

For the CH$_3$ group we note a rather large uncertainty, which arises naturally from the comparably small SSB intensities;

only $\langle P_2 \rangle$ could be estimated with sufficient accuracy. It is not possible to use a smaller spinning speed because of inevitable superpositions with SSBs of other resonances. The accuracy of the oscillation coefficients must be high; otherwise, the uncertainties of the orientational moments become unacceptable. We wonder at this point how spectra like the ones published by Vogt et al. (1990) could be analyzed at all. A precise evaluation would require a rather involved algorithm performing the separation of the many overlapping, differently phased signals, but no comments along this line can be taken from that paper.

One straightforward ambiguity relates to the unknown orientations of the $\delta_{11}$ and $\delta_{22}$ eigenvalues, one of which points along the segmental direction. Results for both options are provided. In all, smaller and more ambiguous CH$_3$-based values suggest that the segment vector definition used here is more convenient than the one using the normal of the dimethyl plane; the latter seems to exhibit more disorder.

## 4    Conclusions

In summary, we could show that our polynomial approximation of MAS spinning sideband intensities provides an efficient approach to extracting chemical-shift tensor elements, with an accuracy that can match dedicated single-crystal experiments or the measurement of static powder lineshapes in single-site isotope-labeled compounds. It is stressed that the approach provides in principle arbitrary accuracy and no specific numerical procedures (such as finite-step integrations). We have provided so far unreported tensor parameters for selected aromatic $^{13}$C resonances of polycarbonate.

The approach is particularly suited for the determination of orientation degrees in anisotropic samples from spinning sidebands taken from 2D syncMAS spectra. Here, a number of so far underestimated fundamental problems was addressed, on the one hand related to phase distortions inherent to the syncMAS method, and on the other hand related to the tensor orientations in the studied sample, i.e., stretched polycarbonate. Based on our polynomial approximation considering terms up to the





6th power in $\delta\omega_0/\omega_r$, we could confirm the results from previous studies of chain orientation in this polymer, but could also
provide a critical perspective and the need for further studies, possibly using isotope-labeled samples to confirm some of the
necessary assumptions.

*Author contributions.* GH designed the research, performed theoretical derivations, experiments and analyses and wrote the paper. KS discussed results and wrote the paper. PS and DRL designed the research concerning orientation effects in stretched polymer glasses, provided the sample and discussed results.

*Data availability.* Experimental data are available upon request from the corresponding author.

*Competing interests.* The authors declare no competing interests.



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
