# Peer review of "Efficient polynomial analysis of MAS spinning sidebands and application to order parameter determination in anisotropic samples"

_Magnetic Resonance, 2021_

## Author Response (AR1)

**Comments and questions of Referee 1, our changes:**

The line numbers in our responses refer to the numbering of the revised version even if the referee comments contain the numbering from the former version.

Comment 1: Even though the samples and experimental conditions are not the major concern of the study, and the information is given somewhere in the manuscript, it is still easier to read if summarized in a dedicated section.

Change in manuscript: In the revised version a section "Experiments" was added. It contains the experimental conditions and parameters.

Comment: In line 487 reads "The second to last column in Table 3 is the noise-related sum squared uncertainty of 10 SSB intensities considered" and in Table 3 the 2nd last column is  $10\Delta I2$ . Does this supposed to mean 10-time the integration uncertainty of some sort of "average SSB", assuming each SSB has similar line width?

Changes in manuscript: In table 3, second-last column, 10  $\Delta I^2$  was replaced by  $(\Delta \mathbf{Y})^2$  because the sum of the absolute uncertainties of the used SSB is meant here.

Line 540: A sentence was changed which is related to the table content.

Comment: Following #2 in line 488 reads "Only for the C1+C2+C5 combined signal does  $\chi^2$  from the best fit exceed this value significantly". From Table 3,  $\chi^2$  of C3 and C4 is even bigger compared to the uncertainty, as discussed in the later section. Did I misunderstand something?

Change in manuscript: The typing errors are removed; the correct exponents were inserted in table 3.

Comment: Besides, there are a few typos to be corrected:

- 1. Line 36, "different order are separated"
- 2. In Fig. 2 caption, should be "for  $\eta=1$ " instead of "for q=1"
- 3. Line 389, "the intensity of this valuable signal"
- 4. In eqn. 40, a missing second half parentheses after  $\omega 2$ .

Change in manuscript: The typos were removed in the revised version.

**Comments and questions of Referee 2, our changes:**

Referee: This paper introduces an exact polynomial representation of SSB intensities to overcome the "far too complicated to be applied", in the words of the authors, Herzfeld-Berger relation.

Change in manuscript: This expression in line 39 was changed in "rather complicated equation (25)".

**Comment: The first question that comes up is the justification that the polynomial evaluation will be appreciably faster than a numerical powder average of the expression that is routinely used.**

Change in manuscript: We added some sentences starting at line 210 in which we describe why the SSB calculation by polynomials will be appreciably faster than by a numerical simulation (simpler structure of the function, powder average already included in polynomials, no Fourier analysis required, use for

estimation of uncertainties). Please note: latexdiff did not mark the newly inserted items (lines 214 ... 224 and lines 229 ... 233), only the text parts next to them.

**Comment: Secondly, it is not clear how the polynomial expression that is obtained is exact, as is claimed; it is an approximation to some order.**

Change in manuscript: In lines 201 to 204, some sentences were added. They state that equation (21) is exact because during its derivation, no approximation were used. Each desired accuracy can be reached if a sufficient number of terms has been used.

Comment: That said, there are some interesting aspects to the paper. Taking sum and difference of positive and negative sidebands of a particular order is one such. Is this only true for polynomial expansion or is it general? So also is the use of symbolic manipulation programs for evaluating the coefficients.

Change in manuscript: In line 246 we added a sentence which points out that the separation of the dependencies is of course not a consequence of this kind of mathematical treatment but it can be visualized and used very easy in the polynomial representation.

Question: It is understandable that in successive rotations the last rotation of the previous rotation and the first rotation of the present can be combined because they are about the same axis. However, wouldn't the first rotation of the first and the last rotation of the last remain?

Change in manuscript: We added a sentence in line 94 explaining that the last rotation is around  $\mathbf{B}_0$  and has therefore no meaning.

Comment: Because of performing rotations in the vector space spanned by the tensors, rotations are treated as a left multiplication with a 5 x 5 matrix. The advantage that this affords over bilinear matrix operations would depend on the number of multiplication and addition operations involved, and not on the size (storage is a minor factor).

Change in manuscript: We added a sentence in line 102 explaining that the advantage of the tensorspace operations consists of a reduction of the number of operations.

Question: In equation 25 (and may be a few others), the coefficients vary over orders of magnitude with some of the coefficients being extremely small. Are these small coefficients really significant?

Comment: In Figure 2, one find that the  $I_+$  intensity for the 12th order polynomial lies between 6 and 12. Increase the order would make it go further towards 2? That looks counterinutitive.

Change in manuscript: A sentence was added to the figure caption which explains that because of alternating signs of terms off different order, the curve alternatively approaches of removes from the curve of second order. This explains at the same time the significance of the small coefficients because they are the differences between the curves of different order.

**Question: In Figure 4, what are the triangles and what is the red curve?**

Change in manuscript: We added the necessary explanation of the symbols to the Figure caption.

Comment: Something is missing from the sentence starting at the end of line 321.

Change in manuscript: This sentence was omitted because experimental parameters are introduced in the new section "Experiments", see above.

**Question: Why is the \delta of C1 and C2 in Table 2 different from that on line 374?**

No change in manuscript, because after checking the manuscript we found that the  $\delta$  values in line 374 are equal to that in table 2.

Comment: Equation 38 assumes that C2 and C5 are polarized identically despite the protons around them being two bonds apart in one case and three bonds apart in the other.

Change in manuscript: We added two sentences below equation (38) where we explain that for both C2 and C5 the next protons are two bonds apart, and therefore it seems to be justified to assume equal CP efficiencies for both carbons.

**Question: What is the origin of equation 39?**

Change in manuscript: Two sentences were added by which we explain the origin of this equation (line 452) and the use as confirmation of the stability in time of the 2D experiment (line 455).

**Other changes**

Equations (43) and (44): All variables get italic style.